# Immunotherapy with PD-1 Inhibitor Nivolumab in Recurrent/Metastatic Platinum Refractory Head and Neck Cancers—Early Experiences from Romania and Literature Review

**DOI:** 10.3390/diagnostics13162620

**Published:** 2023-08-08

**Authors:** Camil Ciprian Mireștean, Mihai Cosmin Stan, Michael Schenker, Constantin Volovăț, Simona Ruxandra Volovăț, Dragoș Teodor Petru Iancu, Roxana Irina Iancu, Florinel Bădulescu

**Affiliations:** 1Department of Medical Oncology and Radiotherapy, University of Medicine and Pharmacy of Craiova, 200349 Craiova, Romania; mc3313@yahoo.com (C.C.M.); mihai_csmn@yahoo.com (M.C.S.); fbadulescu2001@yahoo.com (F.B.); 2Department of Surgery, Railways Clinical Hospital, 700506 Iași, Romania; 3Department of Medical Oncology, Vâlcea County Emergency Hospital, 200300 Râmnicu Vâlcea, Romania; 4Department of Medical Oncology, “Sf Nectarie” Oncology Center, 200347 Craiova, Romania; 5Department of Medical Oncology, Clinical Emergency County Hospital, 200642 Craiova, Romania; 6Department of Medical Oncology and Radiotherapy, “Grigore T. Popa” University of Medicine and Pharmacy, 700115 Iași, Romania; cvolovat@gmail.com (C.V.); dt_iancu@yahoo.com (D.T.P.I.); 7Department of Medical Oncology, Euroclinic Oncology Center, Victoria Hospital, 700110 Iași, Romania; 8Department of Medical Oncology, Regional Institute of Oncology, 700483 Iași, Romania; 9Department of Radiation Oncology, Regional Institute of Oncology, 700483 Iași, Romania; 10Oral Pathology Department, “Grigore T. Popa” University of Medicine and Pharmacy, 700115 Iași, Romania; roxana.iancu@umfiasi.ro; 11Clinical Laboratory Department, “Sf. Spiridon” Emergency University Hospital, 700111 Iaşi, Romania

**Keywords:** head and neck squamous cell carcinoma (HNSCC), Nivolumab, immunotherapy, neutrophil-to-lymphocyte ratio (NLR), second primary malignancy (SPM), platinum refractory

## Abstract

Prognosis in recurrent/metastatic head and neck squamous-cell carcinoma (HNSCC) refractory to platinum-based chemotherapy is poor, making therapy optimization a priority. Anti-programmed cell death protein 1 (anti-PD-1) monoclonal antibody Nivolumab was approved in such cases. We present the early experience with Nivolumab immunotherapy at three cancer clinics from south and northeast Romania, aiming to describe the main characteristics and outcomes relative to literature reports, and to suggest patient selection criteria. Diagnostic, clinical, biological, therapeutic, and outcomes-related data from January 2020 until March 2023 were analyzed retrospectively. Eighteen patients with platinum refractory HNSCC (85.7% men, median age 58.9) were administered Nivolumab for 1–14 months (median 5.6 months) in addition to other treatments (surgery, radiotherapy, chemotherapy), and monitored for up to 25 months. Median neutrophil-to-lymphocyte ratio (NLR) ranged from 2.72 initially to 6.01 during treatment. Overall survival (OS) was 16 months, and patients who died early had the sharpest NLR increases (13.07/month). There were no severe immune-related adverse events. Lower NLR values and combined intensive chemotherapy, radiotherapy, and immunotherapy were related to better outcomes. To our knowledge, we also report the first two cases of second primary malignancy (SPM) in the head and neck region treated with Nivolumab in Romania (for which the sequential administration of radiotherapy and immunotherapy seems better). The work of other Romanian authors on the role of HPV status in HNC is also discussed. Multi-center trials are needed in order to investigate and confirm these observations.

## 1. Introduction

Head and neck squamous cell carcinoma (HNSCC) is associated with poor prognosis and low survival rates in its advanced stages, even if multimodal treatment is administered according to the latest standards. Improved diagnostic and therapeutic approaches have brought minimal benefits to HNSCC prognosis. According to preclinical data, the therapeutic failure of HNSCC can be attributed to genetic mutations accumulating during carcinogenesis and which can elude immunosurveillance, as well as to immunosuppression that accompanies disease progression. In such a context, the identification of strategies to restore the antitumor immune response could improve treatment response and extend survival. 

Nivolumab is one anti–programmed death 1 (PD-1) monoclonal antibody that has been shown to bring long-term overall survival (OS) benefits in the case of platinum-refractory HNSCC, compared to single agent chemotherapy treatment [1]. Scientists are currently seeking to identify predictive biomarkers for such favorable responses to immunotherapy as head and neck tumors known to promote a highly immunosuppressive tumor microenvironment (TME). Programmed death ligand-1 (PD-L1), combined positive score (CPS), tumor mutation burden (TMB), and neutrophil-to-lymphocyte ratio (NLR) are being evaluated individually or in complex models to stratify HNSCC patients and identify who would benefit the most from ICI [1,2,3,4,5,6,7,8]. Concurrently, radiomics, a new field of research based on artificial intelligence applied to medical imaging, promises to contribute meaningfully to this endeavor [4]. 

This study aims to share early experiences of PD-1 inhibitor Nivolumab administration to Romanian patients with relapsed/metastatic head and neck cancers (HNC) refractory to platinum agents. The cases presented and discussed herein were of patients diagnosed and treated in three oncology clinics located in two of the largest academic medical centers in Romania—Craiova (in the south) and Iași (in the north-east).

## 2. Materials and Methods

This is a retrospective analysis of locoregional relapse or metastatic HNC cases of patients treated with Nivolumab during January 2020–March 2023 at three specialized clinics in Romania: the Oncology Clinic of the Craiova County Emergency Hospital; “Sf Nectarie” Oncology Center in Craiova; and Euroclinic Oncology Center of Victoria Hospital in Iași. The study was formally approved by the Ethics Committee of the University of Medicine and Pharmacy of Craiova (no. 125/15.06.2023).

The main inclusion criteria were the aforementioned diagnosis, treatment with Nivolumab, and relevant follow-up information in the patients’ medical records for the purposes of this study. Medical histories of autoimmune disease, malignancy, as well as therapies with immune checkpoint inhibitors other than Nivolumab were considered exclusion criteria. Biological, clinical, and therapeutic data, including outcomes, were analyzed. The sample size was evaluated with the OpenEpi online calculator, and note should be made of the fact that it did not meet the required value for a 90% confidence interval, so the statistical power of the results is limited. 

## 3. Results

Eighteen patients with locoregional relapse or metastatic HNC refractory to platinum-based chemotherapy were started on Nivolumab and subsequently monitored at the aforementioned clinics during the studied period. The patients’ ages ranged from 52 to 65 (median age 58.9 years), and the majority were men (16 cases, 85.7%). 

Except for one case of adenoid cystic carcinoma (5.5%), seventeen patients were being treated for squamous cell carcinomas (94.5%). In five cases (27.5%), the tumors were anatomically located in the oropharynx (the tonsils and base of the tongue), and in another five (27.5%), they were in the oral cavity (the tongue and floor of the mouth). Three patients had laryngeal cancer (16.5%) and another two had sinonasal cancer (11%). There were also one case of hypopharyngeal cancer (5.5%) and one of sinonasal cancer (5.5%), both graded as stage I. In fourteen cases (77%), the tumors had already started to spread: eight patients had stage III cancer (44%), while five were in stage IVa (27.5%), and one was considered stage IVb due to evidence of lung metastases (5.5%).

Treatment wise, two patients benefitted from curative surgical treatment (11%). One patient (5.5%) underwent salvage surgery after relapse (5.5%), and another had surgery for residual disease after experiencing a response to immunotherapy. Radiotherapy was used in most cases (15 cases, 82.5%), either as definitive treatment (11 cases, 60.5%), adjuvant therapy (2 cases, 11%), or re-irradiation (2 cases, 11%). There was one other case where radiotherapy was used for palliative purposes. Total doses ranged between 30 Gy in 10 fractions and 70 Gy in 35 or 33 fractions (simultaneous integrated boost). Poly-chemotherapy or single agent chemotherapy was administered in all cases, with the number of cycles varying between 1 and 10. Chemotherapy regimens included platinum-based drugs Cisplatin, Carboplatin, 5-Fluorouracil, and Docetaxel in combinations of two or three. Where single agent chemotherapy was used, the administered drugs were Capecitabine, Methotrexate, or Carboplatin. 

Immunotherapy was administered for 1 to 14 months, depending on the case (median duration 5.6 months). Four patients (22%) died within three months of Nivolumab initiation. Otherwise, the follow-up period ranged between 3 and 25 months. 

The median disease-free survival (DFS) was 34 months, ranging from 0 to 84 months. Most negative developments were locoregional recurrences, but there were also two cases of lung metastases (11%), one case of mediastinal lymph node metastases (5.5%), and one case of both locoregional and lung recurrence (5.5%) (Table 1 and Table 2). 

Only one patient was diagnosed with metachronous pancreatic cancer. Two patients met the criteria for second primary malignancy (SPM), both cases after more than 10 years of initial disease-free survival (DFS). One of these patients underwent re-irradiation and the overall survival (OS) after initiation of Nivolumab immunotherapy was 16 months. This patient received a total dose of 50 Gy in 25 daily fractions (both initial treatment and re-irradiation were delivered with a Rokus-M40 former Soviet Union Cobalt-60 machine without image guided treatment capability). The other patient with SPM was treated with radiotherapy only for the first malignancy, while the second was treated with intensive chemotherapy and Cetuximab. In this case, the OS after the initiation of immunotherapy was three months. It is also worth mentioning that this patient required corticosteroids and antibiotics after the initiation of Nivolumab. This was one of three cases (16.5%) of complications consisting in bacterial infections with Klebsiella species, Pseudomonas aeruginosa, and/or methicillin-resistant Staphylococcus aureus (MRSA). 

No immune-related adverse events (AEs) of grade 3 severity or worse were recorded. One death following admission to the emergency room was due to respiratory failure considered unrelated to immune-mediated pneumonitis. 

For the male patients, the initial median neutrophil-to-lymphocyte ratio (NLR) for the prostate-specific antigen nadir (NLR-nadir) was 2.72 (range from 1.43 to 8.27), and during the studied treatment period, the median value increased to 6.01 (range from 2.91 to 16.08). The sharpest increases of NLR were observed in the patients who died early after immunotherapy was initiated. The NLR variation in these cases featured positive increments of as much as 13.07 per month.

## 4. Discussions

### 4.1. Immunotherapy in HNC—The Challenges of a Revolution

Locally advanced HNSCC is characterized by a recurrence rate of over 50% within 3 years of diagnosis. In relapsed/metastatic cases that prove refractory to platinum treatment, therapeutic options are limited, and patients do not generally survive longer than six months. The disease progresses rapidly through invasion and metastasis that have been associated with programmed death ligands (PD-L1 and PD-L2) and with programmed cell death Protein 1 (PD-1) expressed on the surface of T lymphocytes. Therefore, the use of immune checkpoint inhibitors has been investigated in multiple preclinical and clinical studies. For example, the PD-1 inhibitor Nivolumab was trialed on 361 patients in a dosage of 3 mg per kilogram every 2 weeks, and outcomes were compared with single-agent of systemic therapy, including treatment with Methotrexate, Docetaxel, or Cetuximab. Overall survival (OS), progression-free survival (PFS), objective response rate (ORR), quality of life, and safety profile were assessed. The median OS of patients treated with Nivolumab was 7.5 months compared to 5.1 months in the case of investigator-chosen monotherapy. Moreover, the treatment response rate was 13.3% versus 5.8% in favor of the patient group treated with immune checkpoint inhibitors (ICI), and the toxicity profile of Nivolumab was significantly superior (13.1% adverse events of grade 3 or 4 versus 35.1%) [1,9,10,11].

In a subgroup analysis, Gillison et al. reported better long-term survival rates for patients with recurrent/metastatic HNSCC who received Nivolumab as first-line treatment. The authors calculated a median 2-year OS of 20.4%, compared to 3.8% in the group that received other, investigator-chosen treatment from the standard options. Thus, immunotherapy as first-line treatment for recurrent or metastatic HNSCC is clearly beneficial. Before the introduction of ICI therapy in recurrent/metastatic HNSCC, some of the preferred options have been the so-called EXTREME regimen consisting of Cetuximab, platinum, and 5-Fluorouracil, and later on, the TPEx regimen including Cetuximab, platinum salts, and Docetaxel [12].

A combined positive score (CPS) > 1 was proposed as an indicator for Pembrolizumab monotherapy instead of Pembrolizmumab in combination with platinum and fluorouracil. Pembrolizumab monotherapy was shown to extend OS compared to the EXTREME regimen for cases with CPS > 1, and similar benefits were noted regardless of CPS values when administered in association with platinum and fluorouracil. The double association between a Cytotoxic T-Lymphocyte-associated protein-4 (CTLA-4) inhibitor, Ipilimumab, and Nivolumab is still under investigation [1,9,13].

In the KEYNOTE-040 trial, a phase-3 trial on 247 patients from 97 medical centers across 20 countries; the combined research teams were able to demonstrate an OS benefit of 1.5 months in the intention-to-treat patient population compared to standard treatment based on Methotrexate, Docetaxel, or Cetuximab. These were patients with metastatic/recurring HNSCC whose platinum-based chemotherapy was administered in association with Pembrolizumab as immunotherapeutic agent [14]. The use of immunotherapy in the first line of treatment for patients with high expression of PD-L1 was anticipated in 2019 by Saada-Bouzid et al., who mentioned the potential of immunotherapy with Pemrolizumab to become first-line treatment in recurrent and metastatic HNSCC, particularly for patients with high PD-L1 levels [15].

In 2021, a meta-analysis and systematic review was published that aimed to assess the efficacy of anti-PD-1 therapy vs. anti-PD-L1 therapy vs. standard chemotherapy in patients with relapsed or metastatic HNSCC. The review considered the results of several trials on a total of three thousand patients, including Keynote 040, Keynote 048, Eagle, Condor, Checkmate 141. The analysis did not highlight major differences between the studied groups in terms of treatment results, but anti-PD-1 administration was associated with better results in smokers and in HPV-negative patients. Anti-PD-L1 was associated with favorable response in female patients and smokers. The results of the meta-analysis could help stratify patients to optimize the choice of immune checkpoint inhibitors (ICI) [16].

A post hoc analysis of patient data from the CheckMate 141 trial evaluated the results of Nivolumab administration in recurrent or metastatic HNSCC cases treated beyond first RECIST-defined progression (TBP). Among the 60 patients who met this criterion, 25% presented a significant reduction in the size of the target lesions. Moreover, 5% of these cases were associated with a reduction in tumor size of over 30%. The authors also pointed out the need to identify biomarkers of response beyond disease progression [17].

Once Nivolumab has been administered to a patient with recurrent/metastatic HNSCC, research results suggest that salvage chemotherapy (SCT) can be a good therapeutic option. In a study on 21 patients receiving SCT after Nivolumab, the overall response rate (ORR) was 52.4%, and the disease control rate was 81.0%. Moreover, the median OS was 12.9 months and the median PFS was 5.4 months. Positive PD-L1 expression could be considered a prognostic factor of response to SCT [18].

The efficacy and safety of Nivolumab for patients with relapsed and metastatic HNSCC sensitive to platinum was evaluated in a single-center prospective study in Japan. Cases were included based on the criterion of relapse or metastasis at least 6 months after therapy or chemoradiotherapy based on platinum salts. In the group of 22 platinum-sensitive patients, the median OS was 17.4 months and the survival at 1 year was 73%. With an ORR of 36% and no adverse events (AEs) of grade 4 and 5 related to treatment, the regimen was considered safe and feasible [19]. Furthermore, the retrospective study by Okada et al. on 88 patients (of whom 60 platinum-refractory cases) yielded similar results for the use of Nivolumab in metastatic or relapsed HNSCC, regardless of the sensitivity or refractory status of the response to platinum-based treatment [20].

Regarding potentially useful biomarkers, Gavrielatou et al. analyzed HNSCC response to ICI treatment and proposed a classification into three categories: tumor-related biomarkers; host-related biomarkers; and other biomarkers related to the tumor microenvironment (TME). The first category features PD-L1 or CPS, Interferon gamma (IFN-γ) signature genes, and tumoral mutation burden (TMB). An “inflamed” microenvironment is an immune environment including increased values of CD3 and CD8 lymphocytes, forkhead box protein 3 (FOXP3), T cell clonality, M1 macrophages, and tertiary lymphoid structures (TLS). The presence of human papillomavirus (HPV-positive status) and a microbiome including Akkermansia muciniphila and Enterococcus hirae have been associated with a positive response to immunotherapy [21].

### 4.2. Neutrophil-to-Lymphocytes Ratio (NLR)—Towards a New Biomarker

The hypothesized potential of the neutrophil-to-lymphocytes ratio (NLR) as an inflammation marker and of the modified Glasgow prognostic score (mGPS) to predict the response to Nivolumab was evaluated in a multi-center Japanese study on 88 patients with HNSCC. Higher pre-treatment NLR and mGPS (=2) were associated with lower OS rates. Thus, a low NLR value was correlated with higher OS at 1 year (45.3%) vs. 16.3% for cases with a high NLR. Moreover, there were significant differences between the 1-year OS of patients with an mGPS of 0 versus those with an mGPS of 2 (37.4% vs. 26.1%) [22]. In another Japanese cohort study, high NLR values and especially a cut-off value of 5 were predictive of unfavorable response justifying the discontinuation of immunotherapy and the switch to “best supportive care” [23]. Several authors proposed the use of NLR as a biomarker for evaluating the effectiveness of immunotherapy with Nivolumab [23,24,25,26].

Despite such results, the association of neutrophils with cancer progression and treatment outcomes is controversial. According to an umbrella systematic review and meta-analysis of a large number of studies published in the MEDLINE, Embase, and Cochrane databases, there was consistent evidence supporting the correlation between NLR and tumor-associated neutrophils (TAN) on one hand, and an unfavorable cancer prognosis on the other hand. However, certain aspects and relationships were still lacking in evidence, including causality and clinical utility, calling for further research [27].

A higher NLR ratio was significantly correlated with an unfavorable response to treatment, overall survival, and reduced progression-free survival in patients treated with ICI for various types of cancer. Using NLR in conjunction with tumor mutational burden (TMB) can increase the predictive power of such information. In one study, for example, low NLR and high TMB were associated with a favorable response to ICI, while increased NLR and low TMB predicted immunotherapy failure [27]. In another study on 140 patients with unresectable or metastatic esophageal cancer, the prognostic role of NLR was evaluated in the context of immunotherapy with anti-PD-1 agents. The study proposed a follow-up period of 20 months and a cut-off NLR value of 5. Pre-treatment NLR values of at least 5 were associated with higher ORR (46.7%), PFS of 10 months, and OS of 22 months. For pre-treatment NLR under 5, ORR was substantially lower (12.1%), as were PFS (3.5 months) and OS (4.9 months). The study thus demonstrated that pre-treatment NLR can be used as a predictive biomarker of treatment response and prognosis in unresectable and metastatic esophageal cancer treated with anti-PD-1 immunotherapy [28]. 

In the context of treating recurrent or metastatic oral squamous cell carcinoma (OSCC), NLR values and Nivolumab treatment outcomes were evaluated in an observational study of 13 cases. The complete and partial response rates were 38.5% and 0%, respectively, and the rates of stable disease and progressive disease in non-responders were 77% and 53.8%, respectively. The study showed a decrease in NLR from a median value of 4.1 to 3.3 in responders. In non-responders, the median NLR increased from 5.6 pre-treatment to 9.4 during treatment. Moreover, post-treatment NLR values higher than 10 were associated with an unfavorable response to post-Nivolumab salvage chemotherapy, and post-treatment NLR higher than 10 significantly predicted unfavorable OS, justifying the authors’ suggestion of using NLR as a predictor of salvage chemotherapy outcomes post-Nivolumab [29].

Other researchers have sought to assess the predictive and prognostic power of NLR and of platelet-to-lymphocyte ratio (PLR), as well as the dynamics of these biomarkers before/during/after treatment with programmed death-ligand 1 (PD-L1) inhibitors as second-line treatment for non-small cell lung cancer (NSCLC). In one study where Pembrolizumab was the inhibitor used, the NLR value of 5 was identified as a cutoff for predicting PFS and OS endpoints. Specifically, patients with pre-treatment NLR above 5 had PFS of 6.86 months, compared to 18.82 months in the cases of patients with pre-treatment NLR up to 5. Moreover, NLR higher than 5 before chemotherapy and the presence of bone metastases were associated with reduced OS. The authors concluded that NLR below 5 could work as a prognostic biomarker of improved survival in NSCLC treated with Pembrolizumab immunotherapy in the second line [30].

The same cut-off value emerged from another study which investigated the dynamics of NLR during the first 21 days of immunotherapy in a group of 509 patients with advanced cancers. A moderate decrease in NLR in the first weeks after the initiation of immunotherapy was associated with the longest OS (median of 27.8 months). Rapid variations, whether they were decreases or increases, were associated with shorter OS (median of 11.4 months) [31]. Non-linear changes in NLR and their correlation with survival were reported for the first time in this study and later confirmed by Li et al. [31,32]. An early decrease in NLR in the first 6 weeks of treatment was associated with good prognosis in patients with metastatic renal cancer treated with immunotherapy. At the same time, a relative change of more than 25% from the baseline NLR value was associated with reduced ORR and OS in the same study [33].

Overall, the ATTRACTION-2 trial demonstrated significant benefits of immunotherapy in the survival of patients with advanced gastric cancer, but different responses in patient subgroups justified more nuanced research on biomarkers and other indicators predictive of treatment response [34]. One such analysis on 98 of the trial patients treated with Nivolumab (3 mg/kg or 240 mg/body every two weeks) found that the following parameters were correlated with treatment response after a median follow-up period of 4.9 months: Eastern Cooperative Oncology Group (ECOG) index values of 0 or 1; pre-treatment NLR higher than 3; and NLR value variations of ≥2 in the first 60 days of treatment. NLR variations in the first 2 months after immunotherapy initiation, specifically, were considered a possible predictive biomarker for treatment with Nivolumab [35]. In a separate analysis by Yamada et al., the NLR value of 2.5 was found useful for defining subgroups of patients according to prognosis in cases of gastric and gastroesophageal junction cancer treated with Nivolumab in monotherapy [36,37]. The research into the potential biomarker value of NLR, but also of the NLR variation during immunotherapy for different types of cancer, is summarized in Table 3 [29,30,31,32,33,36].

### 4.3. Immunotherapy and HNC beyond HNSCC (Focus on Other Histological Types and Second Primary Malignancy)

The potential benefits of Nivolumab have been evaluated in other cancer subtypes not included in the aforementioned CheckMate-141 trial. For instance, Ueda et al. enrolled 97 patients with nasopharyngeal squamous carcinoma and other adenoid cystic carcinomas. While, for nasopharyngeal cancer, the results of the study agreed with those of the NCI-9742 trial and were comparable to historical data, in the case of rare head and neck adenoid cystic and neuroendocrine tumors, only one in fourteen patients had a favorable response, so the authors considered the indication to administer immunotherapy in such cases premature. However, in undifferentiated tumors, Nivolumab proved beneficial [1,38,39].

In a randomized phase-3 study on both relapse and newly diagnosed patients with head and neck squamous cell carcinoma, Patil et al. evaluated the benefit of a low dose of immunotherapy added to triple metronomic chemotherapy (TMC), based on data supporting OS benefits provided by metronomic chemotherapy. The patients had a palliative treatment indication Eastern Cooperative Oncology Group performance status of 0–1. The TMC regimen included oral Methotrexate 9 mg/m^2^ once a week, Celecoxib 200 mg twice a day, and Erlotinib 150 mg once a day. In the experimental group, 20 mg of Nivolumab every 3 weeks was added to TMC, and OS was assessed at 1 year. The compared results showed superior OS in the case of patients treated with both TMC and immunotherapy (10.1 months) relative to the TCM group (6.7 months). Note should also be made of the fact that the rate of adverse events was 3.9% higher in the TMC group. This trial paved the way for a new standard of care for patients with palliative indication that they do not benefit from standard-dose immunotherapy for various reasons [40].

Second primary malignancy (SPM) can be associated with radiotherapy treatment in HNC. It is considered an invasive cancer at a non-contiguous site diagnosed at least 6 months after the completion of radiotherapy. Analyzing the incidence of SPM in the head and neck region in a group of 1512 patients, Ng et al. identified a 9% risk, mainly of oropharyngeal cancer. Growth rates of 4, 10, and 25% were associated with time intervals of 5, 10, and 15 years after completion of treatment, respectively, with an estimated 5-year overall survival of 44%. Smoking was found to be a risk factor not only of HNC, but also of SPM in the head and neck region. However, in the case of hypopharyngeal cancer, SPM is not necessarily considered to decrease OS, and such assertions are methodologically impeded by insufficiently long follow-up periods, according to Guo [41,42]. 

Intensity-modulated radiotherapy (IMRT) is recommended as a technique for re-irradiation, taking into account the limits imposed by possible treatment-related toxicities and the need to reduce the dose to organs at risk (OARs). The risk of oral and thyroid SPM is considered proportional to the radiation dose, and newly published data suggest a reduction of this risk in patients whose first head and neck cancers were treated with ICI. In nasopharyngeal cancer, the use of intensity-modulated irradiation techniques is associated with a significantly lower risk of SPM compared to conventional radiotherapy. An irradiation dose higher than 60 Gy is considered necessary to achieve tumor control, and late toxicity of grade 3 or higher has been estimated at 23.8% [43,44,45].

### 4.4. Radiotherapy and Immunotherapy in HNC—An New Alliance with Promising Perspectives

The synergistic association between immunotherapy and radiotherapy has already gained the attention of both fundamental and clinical researchers. If, in the case of higher doses per fraction, the release of tumor neoantigens is the main pathophysiological substrate associated with the augmentation of the immune response, in the case of irradiation with the standard fractionation regimen, the reprogramming of the tumor microenvironment (TME) and the conversion of a “cold” immune phenotype into an “inflamed” phenotype could be associated with increasing response rates to immunotherapy [46,47,48,49]. 

In HNC, the abundance of tumor infiltrating lymphocytes (TIL) has been associated with a favorable prognosis. The tumor/stroma ratio seems to have an essential role in establishing the prognosis, this proportion being dependent on not only the anatomical location but also on the involvement of viral etiology. The presence of CD8+ T lymphocytes, although reduced in tumors not associated with HPV, has prognostic value. The presence of regulatory T lymphocytes (Treg) has an immunosuppressive role, and a reduced proportion of CD8+ together with an abundance of Treg could induce immunosuppression, limiting the response to immunotherapy. High expression of Foxp3 Tregs is the exception, as it has been associated with CD8 lymphocytes. Moreover, PD-L1 levels have been correlated with a favorable response to the combination of radiotherapy–chemotherapy with Cisplatin, or to the combination of radiotherapy–biological therapy with Cetuxiumab. The presence of natural killer (NK) cells has also been found to predict favorable prognosis, but these cells are less common in HPV-negative HNC. Tumor-associated macrophages (TAM) have been associated with progression, metastasis, and the risk of recurrence after treatment. M2 macrophages are generally associated with the evolution of disease, while M1 type macrophages have an antitumor role. For this reason, TME reprogramming with the help of radiotherapy or chemotherapy with the promotion of M1 macrophages can be an effective strategy to induce tumor regression [48,49,50,51].

The studied synergistic effects of immunotherapy and radiotherapy in HNC were summarized by Wong et al. in six main conclusions. For one, there seems to be a potential for exploitation of the immune landscape characteristic of HNC cancers induced by viruses such as the Epstein–Barr Virus (EBV) in nasopharyngeal cancers and the Human Papilloma Virus (HPV) in oropharyngeal cancers. Currently, the association between immunotherapy and radiotherapy is intensively analyzed in clinical studies, but consensus is yet to be reached once clinical evidence conclusively demonstrates the additional benefits brought by irradiation to immunotherapy [51].

The dual immunosuppressive and immunostimulatory effects of irradiation are in a fragile balance, which needs to be carefully analyzed before transferring results to clinical practice. The sequence, dose, and optimal administration time of each treatment must be adequately gauged in order to maximize the synergistic potential and prevent negative developments. Immunosuppression induced via concurrent radiochemotherapy–immunotherapy, as well as the immunosuppression associated with elective irradiation of lymph nodes, could be the cause of failures reported in clinical trials such as JAVELIN Head and Neck 100 (NCT02952586). Even if it refers to Pembrolizumab, an ICI agent other than Nivolumab, a sequential combination is considered optimal in the case of delivering radio-chemotherapy and immunotherapy [52,53,54,55]. 

Until now, at least nine phase-3 trials have evaluated different immunotherapy agents in a variety of sequences and associations for the treatment of HNC and other cancers. For instance, the CHECKMATE 577 and PACIFIC trials obtained favorable results for immunotherapy administered as maintenance after radiochemotherapy as well as after radiochemotherapy in combination with surgery in lung and esophagus cancers. The IMvoke 010 trial (NCT03452137) explored the outcomes of administering Atezolizumab as maintenance after definitive treatment of any type of HNC with a high risk of recurrence [48,52]. Different sequences and regimens of immunotherapy with Nivolumab are being trialed in induction settings, concurrent settings, or adjuvant/maintenance treatment schemes associated with definitive radiotherapy in standard fractionation, SBRT, or re-irradiation to assess possible desirable synergistic effects (Table 4) [56,57,58,59,60,61,62,63,64,65,66,67].

For some types of cancers, the benefits of ICI treatment are still controversial. Li et al. reported the case of a patient with adenoid cystic carcinoma of the oral cavity and with lung metastases who was treated with low-dose palliative radiotherapy targeting both the tongue and the lungs. The favorable response on both tumor sites suggests a synergistic mechanism engaging the combination of Nivolumab and radiotherapy, and the authors proposed this treatment regimen as a feasible option. However, a phase-2 study on patients with adenoid cystic carcinoma treated with Nivolumab and Ipilimumab did not obtain equally promising overall response, partial response, and complete response, possibly due to the high rate of grade-4 and -5 toxicities observed [67,68].

Unfortunately, in Romania, the routine assessment of HPV status, whether we refer to DNA, RNA testing, or the surrogate marker p16INK4a (p16) tested by immunohistochemistry, is currently not free of charge for patients in the evaluation of the disease. Undoubtedly, in the context of the new eighth edition of TNM staging, this is a source of errors and uncertainties in the design and evaluation of a clinical trial. Even so, it is worth mentioning the results published by Ursu et al., who evaluated the HPV status in HNC patients from NE Romania: only 2 HNSCC cases were found to be driven by HPV out of 189 samples. The authors concluded that a very small proportion of HNC can be attributed to the HPV virus, specifically the HPV 16 subtype, which seems to be the most prevalent in our country [69,70].

## 5. Conclusions

Our patients’ median PFS during immunotherapy exceeded values reported in the literature, and the same holds even in the case of the patient with adenoid cystic carcinoma. This may suggest the possible benefit of Nivolumab related to the nadir NLR in our male patients, which was lower than the cut-off values reported in other studies, but also to the intensive chemotherapy and radiotherapy treatment delivered in most of the cases. The administration of radiotherapy and chemotherapy during the evolution of the disease could be a potentiation factor that augments the immune response by modulation of TME. To our knowledge, we report the first two cases of PMS in the head and neck region treated with Nivolumab in Romania. Lack of access to a more conformal irradiation technique limits the irradiation with maximum doses and also the association of concurrent chemotherapy. The reported data suggest a possible benefit of the sequential administration of radiotherapy, immunotherapy, and SPM. More dynamic variations and, especially, higher increases in NLR could be associated with the poorest outcomes during immunotherapy. 

All study observations including the role of infections and antibiotic treatment, as well as the possible biomarker value of NLR variation during immunotherapy, should be investigated and validated in multicenter clinical trials. The identification of HPV status for oropharyngeal cancers will be essential to generating valid scientific results in future clinical trials including HNC treated with immunotherapy in Romania. 

## Figures and Tables

**Table 1 diagnostics-13-02620-t001:** Disease characteristics (*N* = 18).

Disease Characteristics	Cases (%)
**Histology**	
squamous cell carcinomas (SCC)	17 (94.4%)
adenoid cystic carcinoma	1 (5.5%)
**Anatomical tumor site**	
oropharynx (tonsils and tongue base)	5 (27.5%)
oral cavity (tongue and floor of the mouth)	3 (16.5%)
larynx	3 (16.5%)
hypopharynx	1 (5.5%)
salivary gland	1 (5.5%)
sinonasal	1 (5.5%)
**TNM stage at diagnosis**	
Stage III	8 (44%)
Stage IVA	4 (27.5%)
Stage IVB	1 (5.5%)

**Table 2 diagnostics-13-02620-t002:** Multidisciplinary treatments and patient outcomes (*N* = 18).

Treatment and Outcomes	Cases/Duration (%)
**Surgery**	
curative intent	2 (11%)
salvage	1 (5.5%)
**Chemotherapy**	
median number of cycles	5 (2–10)
**Radiotherapy**	
adjuvant	2 (11%)
definitive	11 (60.5%)
re-irradiation	2 (11%)
palliative	1 (5.5%)
**Immunotherapy**	
median duration (range)	5.6 months (1–14)
**Outcomes**	
median disease-free survival (range)	34 months (0–84)
locoregional recurrence	12 (66.6%)
lung metastases	4 (16.5%)
mediastinal lymph node metastases	1 (5.5%)
locoregional and lung metastases	1 (5.5%)

**Table 3 diagnostics-13-02620-t003:** Overview of research into NLR as a possible biomarker for different cancer types treated with ICI.

Main Objective	Anatomical Cancer Site	Number of Cases	Results/Conclusion	Reference
Evaluation of NLR and TMB as biomarkers for response to ICI	Pan-cancer analysis; 16 cancer types	1714	NLR low/TMB high patient group benefit from ICI therapy	Valero et al., 2021 [28]
Pre-treatment NLR as a predictor biomarker of response to anti-PD-1 agents	Unresectable or metastatic esophageal cancer	140	Higher ORR and longer PFS for NLR < 5; NLR ≥ 5 independent and significant risk of disease progression, lower OS, and poorer response to immunotherapy	Gao et al., 2022 [29]
Prediction of response to second-line Pembrolizaumab therapy using NLR and PLR	Non-small cell lung cancer	119	NLR > 5 before immunotherapy showed significantly shorter mean PFS; NLR > 5 and higher PLR was correlated with poor OS	Petrova et al., 2020 [31]
Biomarker value of NLR dynamic in the first weeks of ICI treatment	Advanced cancers	509	Non-linear change in NLR was correlated with OS	Li et al., 2019 [31]
Pre-treatment and post-treatment NLR evaluation as predictor of outcomes	Lung cancer	2068	Higher NLR; both pre-treatment and post-treatment could predict PFS and OS for cases treated with immunotherapy	Jin et al., 2020 [35]
Correlation of OS and NLR variation during Nivolumab monotherapy	Gastric cancer	98	∆NLR60 ≥ 2 in first 60 days of immunotherapy was correlated with lower OS; a cut-off value of 3 correlated with prognosis in patients treated with Nivolumab in monotherapy	Ota et al., 2020 [36]

**Table 4 diagnostics-13-02620-t004:** Nivolumab and radiotherapy association in clinical trials.

Trial/Phase	Inclusion Population	ICI Treatment Sequence	Status/Reason	Results
NRG HN005 NCT03952585/Phase II/III	HPV+ Oropharynx early stage, non-smokers	Concurrent Maintenance after any definitive therapy	Suspended (scheduled interim monitoring)	Not reported
NCT03539198	Recurrent/metastatic HNSCC with ≥2 metastatic sites	Proton Stereotactic Body Radiotherapy (SBRT) concurrent with ICI	Terminated (failure to accrue)	Not reported
NCT02684253/Phase II	Recurrent/metastatic HNSCC with ≥2 metastatic sites	Stereotactic Body Radiotherapy (SBRT) concurrent with ICI	Completed	No evidence of abscopal effect and no superior response for SBRT + ICI patient group
NCT02764593 RTOG 3504/Phase I	Intermediate or high risk locally advanced HNSCC	Concurrent induction and adjuvant associated with definitive standard fractionation radiotherapy	Completed	Not reported
NCT03317327REPORT/Phase I/II	Recurrent HNSCC after radiotherapy and SPM	Concurrent with re-irradiation and maintenance	Recruiting	Not reported
NCT03247712/Phase I/II	Resectable locally advanced HNSCC	Neoadjuvant, SBRT, surgery, and adjuvant ICI	Active, not recruiting	Not reported
NCT03576417Nivo PostOp/Phase 3	Locally advanced SCCHN with extra capsular extension and/or positive margins	Concurrent maintenance ICI with chemoradiotherapy	Recruiting	Not reported

## Data Availability

Not applicable.

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
