# Peer review of "Immunotherapy with PD-1 Inhibitor Nivolumab in Recurrent/Metastatic Platinum Refractory Head and Neck Cancers—Early Experiences from Romania and Literature Review"

_diagnostics, 2023, doi:10.3390/diagnostics13162620_

Round 1

Reviewer 1 Report (Previous Reviewer 3)

I accept all responses to my previous comments. I think the manuscript is ready for publication

Author Response

Dear reviewer,

Thank you for the appreciation and the effort to evaluate the article. At the suggestion of another reviewer, we introduced inclusion criteria and sample size evaluation, inserts marked in blue in the text. We also reformulated and corrected expressions in English, especially in the discussion chapter. We hope you will appreciate this updated version of the manuscript.

Kind regards,

Camil Mirestean

Reviewer 2 Report (Previous Reviewer 2)

1.The authors need to elaborate on the Inclusion & Exclusion criteria in their materials & methods section as highlighted in the attachment.

2.Quality of writtten english needs further improvement especially in the discussion section.

3.The methodology for sample size estimation has not been mentioned.

Quality of writtten english needs further improvement especially in the discussion section.

Author Response

Dear reviewer,

Thank you for the appreciation and the effort to evaluate the article. At your suggestion, we introduced inclusion criteria and sample size evaluation, inserts marked in blue in the text. We also reformulated and corrected expressions in English, especially in the discussion chapter. We hope you will appreciate this updated version of the manuscript.

Kind regards,

Camil Mirestean

Reviewer 3 Report (Previous Reviewer 1)

none.

Author Response

Dear reviewer,

Thank you for the appreciation and the effort to evaluate the article. At the suggestion of another reviewer, we introduced inclusion criteria and sample size evaluation, inserts marked in blue in the text. We also reformulated and corrected expressions in English, especially in the discussion chapter. We hope you will appreciate this updated version of the manuscript.

Kind regards,

Camil Mirestean

This manuscript is a resubmission of an earlier submission. The following is a list of the peer review reports and author responses from that submission.

Round 1

Reviewer 1 Report

dear authors,

Yours review of several ongoing clinical trials is well done and explains the value of NLR, TMC and other parameters related to progression or response to therapy performed by each study.

But your study does not show them correctly because of many doubts which I explain below.

1) The cases are of different anatomical location, it is good practice to take homogeneous data. Squamous cell carcinoma of the tongue has a different stage and prognosis than HPV-positive oropharyngeal carcinoma.

2) The 2 cases of oropharyngeal carcinoma reported are neither HPV-positive nor HPV-negative. In fact, these cancers are very different in prognosis and treatment approach. 

3) there are no correct NLR values or other data (pathological or biological) well reported and sufficient to justify treatment results.

4) the large number of cases are squamous cell carcinoma; salivary gland cancer, it is well known, does not respond well to immunocheckpoint inhibitors.

5) the cases are few and cannot be statistically evaluated.

in the few cases the statistical variability is very high and not enough to show a real value to agree or disagree with the literature you have written.

With regret for me this work is not suitable for publication in this journal.

Best Regards

Reviewer 2 Report

1.The manuscript is written in a very unprofessional and shabby manner and the flow of the written content is quite poor and incoherent.

2.The compilation of the results and its presentation is extremely poor and needs to be rewritten in a more coherent manner.

3.The quality of written english is extremely poor and needs a major overhaul.

4.Further comments and suggestions are provided in the attached manuscript.

Reviewer 3 Report

The paper reports the study of the early experience of two academic center with the PD-1 inhibitor Nivolumab in relapsed/metastatic head and neck cancers (HNC) refractory to platinum agents. However, data are limited and insufficient evidence made in the manuscript are not support by data presented. I have several comments and suggestions which I believe need to be addressed before the paper should be accepted.

Specific Comments:

1.     Introduction should be rewritten.

2.     If the institutional human research ethics, please provide the relevant protocol number in both academic center the manuscript.

3.     Table 1: no information about gender.

4.     Table 1-4: the quality was not good enough and should be improved

5.     The discussion and conclusion should be rewritten.

6.     Nivolumab immunotherapy in platinum-refractory HNC is not clear, Clarification of this point in text is needed.

7.     There are some minor grammatical errors.